# Genetic Diversity and Selection Signatures of Lvliang Black Goat Using Genome-Wide SNP Data

**DOI:** 10.3390/ani14213154

**Published:** 2024-11-03

**Authors:** Ke Cai, Wannian Wang, Xu Wang, Zhixu Pang, Zhenqi Zhou, Lifen Cheng, Liying Qiao, Qiaoxia Liu, Yangyang Pan, Kaijie Yang, Wenzhong Liu, Jianhua Liu

**Affiliations:** 1Department of Animal Genetics, Breeding and Reproduction, College of Animal Science, Shanxi Agricultural University, Jinzhong 030801, China; 18574385119@163.com (K.C.); wannian1876@163.com (W.W.); wangxupvt@163.com (X.W.); pang_z_x@163.com (Z.P.); zq204907@163.com (Z.Z.); liyingqiao1970@163.com (L.Q.); panyy@sxau.edu.cn (Y.P.); kjyang@sxau.edu.cn (K.Y.); lwzsxau@gmail.com (W.L.); 2Shanxi Animal Husbandry Technology Extension Service Center, Taiyuan 030001, China; chlf11@139.com (L.C.); zzzlqx@126.com (Q.L.); 3Key Laboratory of Farm Animal Genetic Resources Exploration and Precision Breeding of Shanxi Province, Jinzhong 030801, China

**Keywords:** Lvliang black goat, genetic diversity, family composition, population structure, selection signal

## Abstract

Lvliang black goat (LBG) is an important local goat breed genetic resource in China. At present, the genetic mechanism and germplasm characteristics of LBG have not been fully elucidated, which limits the effectiveness of its conservation and breeding work. This study used genome-wide SNP data to assess the genetic diversity of the LBG population and to identify the relationships between individuals to evaluate their conservation effects. Then, the phylogenetic status of the LBG population was determined by population structure analysis, and its unique genetic structure was found. Finally, we also looked for selection signals in the LBG population and obtained multiple functional terms. The results of this study provide theoretical and practical basis for the protection of LBG genetic resources and contribute to further research on goat germplasm characteristics.

## 1. Introduction

Goat (Capra hircus) is famous for its compact size, low requirements for management and maintenance, and adaptability to various extreme environments. Under natural grazing conditions, goats have an efficient foraging ability, which effectively reduces the economic cost of breeding, making them a preferred breed for low-income farmers [1]. In addition, goats are a versatile livestock that provides humans with critical meat, dairy products, and leather resources, making important contributions to global agricultural systems and local economies [2]. In the Lvliang Mountain area of Shanxi Province, the Lvliang black goat (LBG) has become a multi-purpose local breed after long-term natural selection. It has a medium physique, a rectangular body, black fur, delicious meat, and strong adaptability [3]. In 1979, the number of LBGs in Shanxi Province accounted for about 40% of the total number of goats in the province. However, due to the extensive use of LBG in breeding with other breeds during the development of new varieties, the number of purebred LBGs has dropped sharply. In a 2009 survey of purebred LBGs, only about 440 were found. The LBG was once on the verge of extinction. Therefore, LBG, an important local goat breed resource, has received attention from the country, and an LBG conservation farm was established in 2010 to carry out conservation work.

With the continuous advancement of high-throughput sequencing technology and the significant reduction in costs, genome analysis tools such as whole-genome resequencing technology and single nucleotide polymorphism (SNP) chips have gradually become important to study populations’ genetic diversity and structural characteristics. Genetic diversity is an important component of biodiversity and is crucial to the adaptability and long-term survival of species. Illumina Goat SNP50 BeadChip (San Diego, CA, USA) was used to evaluate the genetic diversity of 169 goat breeds worldwide and found that breeds that were more affected by geographical isolation and strict breeding management measures generally had lower heterozygosity, indicating that the gene flow of these breeds may be restricted [4]. Population structure reflects the complexity of historical events experienced by a population in the course of evolution and its adaptation to the environment. By using SNP chip technology to evaluate the population structure of a specific species, it is possible to reveal the domestication history, migration path, and important events in gene mixing of the species in different regions, while identifying clustering trends within the population and determining the genetic origins of important populations [5]. A study using SNP chip technology found that most local populations in South Africa have gene exchange with commercial goat populations and have a common genomic ancestor, which is related to the preference of people in the region to breed goat breeds with good production efficiency. This study reminds policymakers to pay attention to the impact of hybridization and emphasizes the importance of protecting the core population of goats [6]. In a study of Italian goat breeds, it was found that most northern breeds experienced long-term geographical isolation, which resulted in reduced gene flow and exhibited significant genetic clustering trends. The study also pointed out that historical and humanistic factors have an impact on the location in southern Italy and the genetic makeup of the island’s goat breeds [7].

Selection signals refer to the genetic imprints left in the genome of a population by natural and artificial selection. These signals can effectively reveal the germplasm characteristics of a population when the population size is limited or the phenotypic data are incomplete. In an analysis of selection signals of 30 BoHuai goats, key genes related to lipid metabolism and disease resistance were successfully identified [8]. Similarly, analysis of the Pakistani Teddy goat population also identified multiple candidate genes related to important production traits [9]. These studies not only help us understand their unique advantages but also provide a scientific basis for the protection and breeding of target varieties. Selection signal analysis is also applicable when screening genes related to special traits. For example, genes related to environmental adaptability (*DNAJC18*, *HSPA9*, and *SLC23A1*) were found in Canarian goats, which effectively explained the genetic mechanism of this breed’s adaptation to extreme environmental conditions [10].

Determining the genetic diversity and family structure of endangered species will help to build effective conservation and breeding programs, thereby ensuring the continued existence of genetic resources [11]. Analyzing the selection signals of a population can identify genes related to important economic traits and allow us to deeply analyze the characteristics and molecular mechanisms of the population. However, as an important genetic resource of Chinese goats, the genetic mechanism and germplasm characteristics of LBG have not been fully elucidated, which limits the effectiveness of conservation and breeding work.

In this study, we used the Goat 65K SNP chip to explore the kinship and genetic distance between individuals in the LBG core population and to evaluate its conservation effect. By integrating 12 goat populations, the genetic relationship between LBG and multiple goat breeds was revealed. Finally, a variety of selection signal analysis methods were used to explore the genes related to different traits and further analyze their molecular impact on the goat genome. The results of this study provide a theoretical and practical basis for the protection of LBG genetic resources and contribute new strength to further research on goat germplasm characteristics.

## 2. Materials and Methods

### 2.1. DNA Sample Preparation and SNP Genotyping

We selected all male goats (21) and some female goats (27) from the LBG core conservation population, producing a total of 48 LBGs for venous blood sample collection. Goat blood DNA was extracted using the CWE9600 Magbead Blood DNA Kit of Beijing Kangwei Century Biotechnology Co., Ltd. (Beijing, China). The concentration of DNA was accurately measured by Qubit fluorometer, and the purity of DNA samples was detected by NanoDrop micro-spectrophotometer (the accepted value to include the sample in the analyses was OD260/OD280 = 1.7–2.1). Subsequently, 0.8% DNA agarose gel electrophoresis was performed to detect the degree of DNA sample degradation. When the DNA was intact and not contaminated by RNA, the Goat_IGGC_65K_v2 Illumina HD SNP chip was used for genotyping, and, finally, 59727 SNP sites were detected (DS-1).

### 2.2. Data Collection and Preprocessing

We downloaded 50K Goat SNP chip data of 348 individuals from the Goat Genome Database (https://www.goatgenome.org/datasets.html, accessed on 25 July 2024). These data cover 11 breeds, including 6 Chinese goat breeds (Arbas cashmere goat (AC, n = 59), Guangfeng goat (GF, n = 24), Jining blue goat (JN, n = 39), Luoping yellow goat (LP, n = 24), Nanjiang goat (NJ, n = 23), and Qinggoda goat (QG, n = 24)), three commercial goat breeds (Saanen goat (SAA, n = 47), Boer goat (BOE, n = 48), and Angora goat (ANG, n = 46)), and two wild goat breeds, including Eurasian goat (WG1, n = 4) and Cretan goat (WG2, n = 10) (Appendix A).

### 2.3. Genotype Data Quality Control

We extracted the SNP sites common to the two chips and obtained a total of 43,171 SNP sites (DS-2). We used PLINK v1.9 software [12] to perform quality control on the DS-1 and DS-2 datasets. The commands “—geno 0.1”, “—maf 0.01”, and “—mind 0.1” were used to remove SNP sites with a detection rate ≤ 90%, a minimum allele frequency (MAF) ≤ 1%, and a genotype loss rate greater than 10%, respectively. The command “—chr 1–29” only retained SNP sites on autosomes. The remaining SNP sites after quality control were used for subsequent analysis.

### 2.4. Genetic Diversity Analysis

We used PLINK v1.9 to assess the genetic diversity of the LBG population. Specifically, we further used the parameters “—hwe 0.000001” and “—me 0.05 0.1” to exclude sites that deviated from Hardy–Weinberg equilibrium and Mendelian error in DS-1 before conducting the analysis. The observed heterozygosity (Ho) and expected heterozygosity (He) were then estimated using the “—hardy” parameter, and we used the “—freq” parameter to calculate the minimum allele frequency (MAF). In addition, to assess the degree of inbreeding, we calculated the lengths of the continuous homozygous fragments (ROH) using the “—homozyg” parameter and calculated the homozygosity-based inbreeding coefficient (F_ROH_) based on the ROH. Afterwards, we used PLINK v1.9 software to perform linkage disequilibrium screening on the SNP data with the parameter “-indep-pairwise 50 10 0.2”. For the 26,913 SNPs remaining after screening, we used SNeP v1.1 software [13] to estimate the effective population size (*Ne*) of the historical generations of the LBG population, and we used the linkage disequilibrium method of NeEstimator v2.1 software [14] to estimate *Ne* of the current generation [15].

### 2.5. Genetic Structure and Family Composition of LBG Conservation Population

To clarify the families’ composition in the LBG population, the “—mds-plot 4” command of PLINK v1.9 was used to calculate the identity by state (IBS) and genetic distance between individuals in DS-1. Then, the GCTA v1.94.1 software [16] was used to construct the genomic relationship matrix (*G* matrix) [17] to analyze the relationship between individuals. The neighbor-joining method (NJ) [18] was then used to cluster the individuals. Finally, the LBG population was divided into families based on the clustering results of male goats and the criterion that individuals with a kinship greater than 0.1 belonged to the same family.

### 2.6. Population Structure Analysis of 12 Goat Breeds

To clarify the relationship between LBG and the other 11 breeds, the “-indep-pairwise 50 10 0.2” parameter in PLINK v1.9 was used to remove sites with higher LD in DS-2, and the remaining SNP sites were used for population structure analysis. First, PCA analysis was performed to identify the genetic relationship between breeds. Then, ancestral component analysis (K = 2~20) was performed using ADMIXTURE v1.30 [19] to verify the clustering patterns among populations and infer the possible sources of genetic variation in the test populations. In addition, the genetic distance matrix was calculated using the “-distance-matrix” parameter in PLINK v1.9, and the NJ phylogenetic tree was constructed using MEGA v11.0.13 software [20]. Finally, to explore the genetic exchanges among different populations during their formation, WG1 was used as a root to infer potential migration events in Treemix v1.13 software [21]. Assuming that the potential number of migrations between populations, m, was 1 to 10, the R package “OptM v0.1.8” was used to determine the optimal number of migrations [22].

### 2.7. Selection Signals Analysis

To detect the selection signal in the LBG population, we constructed three datasets (DSs). These were DS-A, including the LBG population; DS-B, including the LBG and six Chinese goat populations; and DS-C, including LBG and three commercial goat populations.

Two complementary analytical approaches—the integrated haplotype score (iHS) and the composite likelihood ratio (CLR)—were used to analyze the selection signal within the LBG population (DS-A). Specifically, iHS first used Beagle v5.2 software [23] to fill in missing genotypes and infer the haplotypes of each chromosome. The iHS score for each autosomal SNP locus was calculated using Selecan v2.0.2 software [24], with a maximum spacing of 800,000, and then normalized. Finally, the genome was divided into non-overlapping windows of 100 kb, and the average |iHS| value of each window was calculated as the test statistic. The windows with the top 5% |iHS| values were defined as candidate regions. The CLR value was calculated using SweeD v4.0.0 software [25], with a step size of 20 kb and a window size of 100 kb. The maximum CLR value of each window was used as the test statistic, and the windows with the top 5% CLR values were defined as candidate regions.

To evaluate the characteristics of selection between LBG and different populations, we used two complementary methods, namely the index of genetic differentiation between populations (F_ST_) and selection scanning analysis based on population haplotypes (XP-EHH), to analyze the selection signals of DS-B and DS-C. Specifically, F_ST_ was calculated using VCFtools v0.1.16 software [26], and XPEHH first needed to use Beagle v5.2 software to fill in the genotype data of the target population and the reference population and infer the haplotype. Then, the XPEHH score of each autosomal SNP site was calculated using Selecan v2.0.2 software. After sorting, the 50 kb genomic intervals upstream and downstream of the top 5% SNP sites were identified as candidate regions.

### 2.8. Identification and Functional Annotation of Candidate Genes

The selected genomic intervals were mapped to the goat reference genome Genome assembly ARS1.0 in the Ensembl database (https://ftp.ensembl.org/pub/release-112/gff3/capra_hircus/Capra_hircus.ARS1.112.chr.gff3.gz, accessed on 10 August 2024), and the corresponding genes and their functional annotation information were identified. We used DAVID (https://david.ncifcrf.gov/, accessed on 30 August 2024) online software to perform Gene Ontology (GO) and Kyoto Encyclopedia of Genes and Genomes (KEGG) analysis on the possible selected genomic regions, and Fisher exact test *p* value of 0.05 was used as the threshold for defining significantly enriched functional terms. Figure 1 shows the workflow of this study.

## 3. Results

### 3.1. Genetic Diversity and Family Composition of LBG Population

After quality control of DS-1, 51,222 SNPs were retained. Through genetic diversity analysis, the average Ho and average He of the LBG population were 0.385 and 0.379, respectively. The average inbreeding coefficient based on ROH was 0.024 ± 0.007, and no homozygous fragments were detected in 13 goats. The MAF range was 0.010~0.500, and the distribution was relatively uniform. The largest proportion was the 0.4~0.5 group (27.03%), and the smallest was the 0.1~0.2 group (10.93%) (Figure 2A). The *Ne* of the LBG conservation population in 2024 was estimated to be 119 based on the linkage disequilibrium method. It was also found that the effective population size gradually decreased with the decrease in generations (Appendix A). The effective population sizes of the LBG populations 13 and 98 generations ago were 192 and 1161, respectively.

In order to analyze the relationship between LBG individuals, we constructed the IBS and the *G* matrix. Specifically, the IBS distance values of the LBG population ranged from 0.1824 to 0.3269, and the average genetic distance was 0.2941 ± 0.0453 (Figure 2B). The *G* matrix results showed that the LBG individuals had a medium to low degree of relationship (Figure 2C). Subsequently, the 48 individuals were clustered by the neighbor-joining method, the NJ tree between individuals was constructed, and the LBG population was divided into 13 families (Figure 2D and Appendix A).

### 3.2. Population Structure Analysis

To investigate the genetic clustering patterns of LBG and 11 goat breeds, we collected multiple goat breeds from different geographical locations (Figure 3A). After quality control of DS-2, 37,882 SNPs were obtained for subsequent analysis. In principal component analysis, the three principal components explained 8.38%, 5.47%, and 3.22% of the variation in the whole-genome data, respectively (Figure 3B). The seven Chinese goat populations, including LBG, can be clearly distinguished from the other five goat populations. The LBG population is closest to the NJ and QG populations in the PCA graph, followed by the AC population.

ADMIXTURE software was further used to quantify the degree of admixture between populations. When K was 2, it was observed that the seven Chinese goat breeds were clearly distinguished from the other five foreign goat breeds, but, except for the LP group, the Chinese goat breeds were mixed with a small amount of the genetic background of foreign goat breeds; when K was 4, the LBG, NJ, and QG groups showed similar mixing component ratios, and only when K was 11 did LBG show a different mixing component ratio from the other groups (Figure 4A).

In order to analyze the genetic relationship between different individuals, we constructed the NJ tree based on the genetic distance between individuals (Figure 4B). In the NJ tree, the seven Chinese goat breeds and the other five goat breeds were located at the two ends, which was consistent with the results of PCA analysis. LBG and LP are closest and are grouped together.

We used the Treemix software to analyze the gene exchange between LBG and other goat breeds. When the migration events were 2–10, it was found that the LP population had gene exchange with the LBG population (Appendix A). The log-likelihood value tended to be stable when the number of migrations was four, and the Δm value reached a high peak. Therefore, the optimal gene exchange route model was when the number of migrations was equal to four (Figure 4C). In this optimal migration model, except for the migration weight value of SAA to JN (0.063) with gene exchange less than 0.1, it was observed that the BOE population had gene exchange with the GF and JN populations, and the LP population had gene exchange with the BOE and LBG populations (Figure 4D).

### 3.3. Selection Signals Analysis

We first applied iHS and CLR methods to detect selection signals in DS-A. At the top 5% threshold, the iHS method screened out 1128 candidate genomic regions (Appendix A) and annotated 1615 genes. The three candidate regions with the highest |iHS| scores were located on chromosomes 1, 10, and 19, respectively (Figure 5A), of which the *RPTOR* gene was annotated on this candidate region of chromosome 19, and the *MYO9A* gene was annotated on this candidate region of chromosome 10. In addition, 1173 candidate regions were identified using the CLR method, with a total of 1928 genes annotated (Appendix A). The three candidate regions with the highest CLR values were located on chromosomes 11, 12, and 14, respectively (Figure 5B), of which the *BIRC6* gene was annotated on this candidate region of chromosome 11, and the *DIAPH3* gene was annotated on this candidate region of chromosome 12.

To find the unique selection features of LBG, we applied F_ST_ and XPEHH methods to detect the two datasets, DS-B and DS-C, and divided the candidate regions with a threshold of the top 5%. The F_ST_ method retained 2069 candidate regions in the two datasets (Appendix A), and 2423 and 2784 genes were annotated in the candidate regions of the two datasets, respectively. The top three candidate regions with the highest F_ST_ values in DS-B were located on chromosomes 6, 10, and 14, respectively (Figure 5C), and only the *NEDD4* gene was located in this candidate region of chromosome 10. The top three candidate regions with the highest F_ST_ values in DS-C were located on chromosomes 1, 2, and 6, respectively (Figure 5D), among which the *LOC106502907* gene was annotated in this candidate region of chromosome 2, and the *FAM47E*, *SCARB2*, and *LOC102177148* genes were annotated in this candidate region of chromosome 6. Using the XPEHH method, 1805 candidate regions were screened in DS-B (Appendix A), and a total of 1529 genes were annotated. The top three candidate regions with the highest XPEHH scores were located on chromosomes 1, 17, and 18, respectively (Figure 5E). The *LSAMP* gene was annotated in this region of chromosome 1, the *TMEM132D* gene was annotated in this region of chromosome 17, and the *LOC102180550* and *LOC102180813* genes were annotated in this candidate region of chromosome 18. A total of 1938 candidate regions were screened in DS-C (Appendix A), and a total of 1552 genes were annotated. The top three candidate regions with the highest XPEHH scores were located on chromosomes 9 and 24, respectively (Figure 5F), and no genes were located within this candidate region.

### 3.4. Gene Annotation and Function Enrichment Analysis

We used iHS and CLR methods to jointly identify 310 candidate genes (Appendix A) in DS-A (Figure 6A). In addition to genes related to economic traits, such as growth traits (*CFL2*, *INSIG1*, etc.), production traits (*PTK2*, *RNASEK*, etc.), and reproductive traits (*NLRP14*, *PRLR*, etc.), genes related to appearance traits (*AGO2*, *SPAG17*, etc.) were also identified. Afterwards, we performed GO and KEGG enrichment analysis on the candidate genes and annotated them according to molecular function, cellular component, and biological process (GO classification) (Figure 6B). The results (Appendix A) showed that the 311 candidate genes were mainly enriched in 20 significant items (*p* < 0.05), including “Roof of Mouth Development”, “Bile Acid Binding”, “Progesterone Metabolic Process”, “Androsterone Dehydrogenase Activity”, “Ketosteroid Monooxygenase Activity”, “Daunorubicin Metabolic Process”, and “Doxorubicin Metabolic Process”.

DS-B identified 248 identical candidate genes under the F_ST_ and XPEHH methods, some of which were related to economic traits, including growth traits (*CAMKK2*, *MEF2C*, *RAVER2*, *SGCD*, etc.), production traits (*PPP2R2B*, *PRKG1*, *AGPAT4*, etc.), reproductive traits (*EPHA6*, *CRIM1*, *IZUMO3*, etc.), behavioral patterns (*GRM5*, *QRFP*, etc.), and immune response (*C4BPA*, *FUT8*, *TMEM154*, etc.). Function enrichment analysis showed that 248 genes were mainly enriched in 24 significant items (*p* < 0.05), including “Neuron Differentiation”, “Regulation of Synaptic Transmission, Glutamatergic”, “Neuroepithelial Cell Differentiation”, “Response to Fluid Shear Stress”, “Mucin Type O-Glycan Biosynthesis”, and “Negative Regulation of Vascular Associated Smooth Muscle Cell Proliferation”. Among the candidate genes identified by DS-C, 168 genes were shared, including a series of genes related to growth traits (*SCD*, etc.), production traits (*AGPAT4*, *PKLR*, etc.), immune response (*C9*, *ZNF283*, etc.), and appearance traits (*EDN3*, *ERG*, *NRG3*, *RSPO2*, etc.). These candidate genes were mainly enriched in “positive regulation of canonical Wnt signaling pathway”, “hair follicle maturation”, “response to xenobiotic stimulus”, “extrinsic component of plasma membrane”, “extrinsic component of endosome membrane”, “protein binding”, “RNA polymerase II cis-regulatory region sequence-specific DNA binding”, “DNA-binding transcription factor activity, RNA polymerase II-specific”, “calcium-activated potassium channel activity”, “Cell adhesion molecules”, and “Metabolic pathways”. Among all the annotated candidate genes, 12 genes, including *DLG2*, *GPC5*, *TRNAC-GCA*, *TRNAE-CUC*, *TRNAE-UUC*, *TRNAG-CCC*, *TRNAG-GCC*, *TRNAG-UCC*, *TRNAS-GGA*, *TRNAV-CAC*, *TRNAW-CCA*, and *ZBTB20*, were identified (Figure 6A).

## 4. Discussion

Analyzing population genetic diversity is of great significance for genetic resource assessment. SNP genotyping has been widely used in many livestock animals as an effective means to explore genetic diversity. LBG has a history of nearly a thousand years of breeding. It is a unique genetic resource among Chinese goat breeds and has great value for protection and development. Based on the SNP data of the LBG conservation population, we found that the Ho and He of its conservation population were close, and the population conformed to the Hardy–Weinberg equilibrium. The average MAF of the LBG population was 0.294, which was at a medium level among the six Chinese goat populations (0.234~0.314) [27], and the SNP loci with MAF between 0.4 and 0.5 accounted for the largest proportion. The above results show that the LBG population has high genetic diversity. In addition, the *Ne* of the LBG population in this study showed a downward trend with the decrease in the number of generations, and it may have experienced a population bottleneck in the past, which is consistent with the actual situation.

The breeding population in the breeding farm is usually in a closed breeding state, resulting in a high degree of inbreeding. This high degree of inbreeding may significantly reduce the genetic diversity of the population and change its genetic structure, which may, in turn, affect the health and adaptability of the population [28]. The inbreeding coefficient based on ROH is more accurate when directly estimated using genomic information than the inbreeding coefficient that relies on pedigree information [29]. The F_ROH_ of the LBG group was 0.024, which was at a relatively low level among goat groups (0.017~0.086) [30]. At the same time, by further constructing the IBS and G matrices, we found that the kinship and genetic distance between LBG individuals were relatively small. This low inbreeding degree and kinship in the LBG population are conducive to maintaining the long-term survival and reproduction of the LBG population [31]. We divided the LBG breeding population into 13 independent family lines based on the clustering results of male goats and the relationship between individuals to provide a reference for subsequent breeding work.

The LBG population has a unique phylogenetic position in the genetic resources of goat breeds and can be clearly separated from other goat populations. We integrated 12 goat populations to explore the unique genetic structure of the LBG population and the genetic relationship between multiple goat populations. This study found that Chinese goats can be clearly distinguished from foreign goat populations. Among Chinese goat populations, the LBG population has the closest genetic correlation with the NJ and QG populations, and the genetic clustering is obvious. Combined with admixture analysis, the three populations have similar ancestry ratios, suggesting that they may have a common ancestor, which is similar to previous research results [32]. When K is 11, the LBG population forms its own unique ancestry composition. The genetic continuity of Chinese goats can be traced back to the Eastern Fertile Crescent in the late Neolithic period, showing genetic differentiation between the north and the south, with the southern goat gene pool being closer to ancient goats [33]. The LP population, located near the southwestern border of China, may have retained some genetic characteristics of ancient goats due to the geographical isolation of the Qinghai–Tibet Plateau and the Hengduan Mountains [34]. Our study found that the genetic distance between the LP population and the LBG population is close, and there are signs of gene exchange, so we can speculate that the LBG population also retains some ancient genetic characteristics. In addition, population mixing caused by human migration, trade activities, and cultural exchanges in history may make the genetic composition of the LBG population more complex. Therefore, the LBG population may be a special population that integrates local genetic characteristics and foreign gene flow, which is of great value for studying the domestication history and genetic diversity of Chinese goats.

It is necessary to clarify the germplasm characteristics of the LBG population for breed breeding and protection. Selection signal analysis can improve the efficiency and accuracy of analyzing livestock and poultry germplasm characteristics when the population size is small or there is a lack of phenotypic information. We detected multiple candidate genes related to economic traits in the LBG population through four selection signal analyses. Among them, the protein encoded by the *CFL2* gene belongs to the actin depolymerization factor family, which plays a key role in the function and morphology maintenance of muscle cells. During the differentiation of bovine myoblasts, *CFL2* affects the fusion of myogenic cells and the generation of muscle fibers by precisely regulating the polymerization and depolymerization of actin, thereby significantly affecting muscle development and function [35]. As an important regulatory mediator in the process of myogenic differentiation, *CFL2* can also upregulate the expression of myogenic marker genes such as *MYOD*, *MYOG*, and *MYH3* [36]. The *SCD* gene affects the saturation of fatty acids through the fatty acid desaturase it encodes. It not only directly determines the fatty acid composition of meat but also indirectly affects the comprehensive sensory evaluation and nutritional value of meat quality through the interaction with vitamin A [37]. The protein encoded by the *NLRP14* gene belongs to the NLR family, which plays a key role in the innate immune and reproductive systems of mammals, especially in oocytes [38]. *NLRP14* may affect the development of early embryos and the interaction between embryos and maternal tissues, which is crucial for the successful implantation and subsequent development of embryos [39]. Through functional enrichment analysis of genes in the selected regions, we found that GO and KEGG pathways related to functions such as metabolism and reproduction were generally enriched in different selection signal methods, indicating that the genomic regions in the LBG genome related to germplasm characteristics, such as body growth and development, fat deposition, and reproduction, were affected by positive selection.

On this basis, we also focused on immune traits as an important aspect of evaluating germplasm characteristics. This study identified a series of immune-related candidate genes in the LBG population, including *C4BPA*, *FUT8*, *TMEM154*, *C9*, *ZNF283*, and *PRNP*. *C4BPA* is an immune gene that plays an important role in lipid metabolism in low-fat bovine mammary epithelial cell lines [40] and plays a key role in immune response by targeting the *TLR-4/NF-κB* pathway and factors in the complement and coagulation cascade pathways [41]. In addition to the *C4BPA* gene, the other five candidate genes play key functional roles in the occurrence and development of specific diseases. *FUT8* plays an important role in the infection process of Escherichia coli, and its low expression helps to enhance the resistance of piglets to Escherichia coli [42]. *TMEM154* has been shown to be associated with reduced susceptibility to peste des petits ruminants virus [43] and ovine progressive pneumonia virus in sheep [44]; *C9* plays an important role in the immune response to sheep mastitis [45]; *ZNF283* is involved in the interaction between foot-and-mouth disease virus and the host [46] and can inhibit the production of porcine reproductive and respiratory syndrome virus and the synthesis of viral RNA [47]; and *PRNP* is associated with resistance to scrapie in sheep [48]. It can be seen that these immune genes may be related to the strong disease resistance and adaptability of LBG in the semi-pastoral and semi-stalled breeding mode in hilly and mountainous areas.

In addition, to further explore the genome of LBG, we compared the genetic differences between LBG and commercial goat breeds and found genes associated with physical characteristics, including *ERG*, *NRG3*, *RSPO2*, and *EDN3*. Specifically, *TMPRSS2:ERG* gene fusion has been reported to be involved in bone metabolism at the cellular level by regulating the expression of *ALPL*, *COL1A1*, and *ET-1* [49] and is associated with the horned/polled phenotype of goats [50]. The epidermal growth factor-like domain of NRG3 binds to the ERBB4 receptor as a ligand, affecting the growth and differentiation of chondrocytes [51], and plays a role in bone formation and angiogenesis through interactions with proteins such as EGFR [52]. *NRG3* is also associated with angular limb deformity [53]. Mutations in the *RSPO2* gene can lead to interrupted limb development in animals, causing limb deformities [54]. *EDN3* affects melanocyte proliferation and differentiation through type B endothelin receptor [55]. Its expression level in the skin of black sheep is higher than that in white sheep [56]. Although *EDN3* does not play a major role in sheep pigmentation [57], its role in LBG still needs further study due to differences between species. In this study, there were 12 common genes among the genes annotated by the four selection signal methods, of which 9 were important transfer RNAs involved in protein synthesis, and the other 3 genes (*DLG2*, *GPC5*, and *ZBTB20*) are involved in synaptic transmission and neuronal connection, cell division, growth and migration regulation, and regulation of cardiomyocyte contraction, respectively [58,59,60,61]. These results are helpful to explore the germplasm characteristics of LBG and provide important clues for further research on its genetic background and adaptability.

Limitations of this study include the following: first, the LBG population tested in this study is small and needs to be verified in a larger population. Second, the density of the SNP chip is low and cannot cover the entire genome, so it is necessary to conduct in-depth exploration of LBG through whole-genome sequencing. In addition, the SNP chip used in this study was developed based on cosmopolitan breeds and is not specific to the LBG breed. Finally, we will focus on exploring the genetic mechanisms of the LBG population at the multi-omics and molecular levels.

## 5. Conclusions

LBG has high genetic diversity. We divided the breed into 13 families, the individuals of which were distantly related. The conservation effort was effective. We clarified the phylogenetic status of LBG; identified genes under selection, such as *CFL2*, *SCD*, and *NLRP14*, that were related to various economic traits; and explored the regulation of these genes. This provided data to support the future protection and improvement of LBG and provided a new perspective for improving the genetic diversity of goat populations.

## Figures and Tables

**Figure 1 animals-14-03154-f001:**
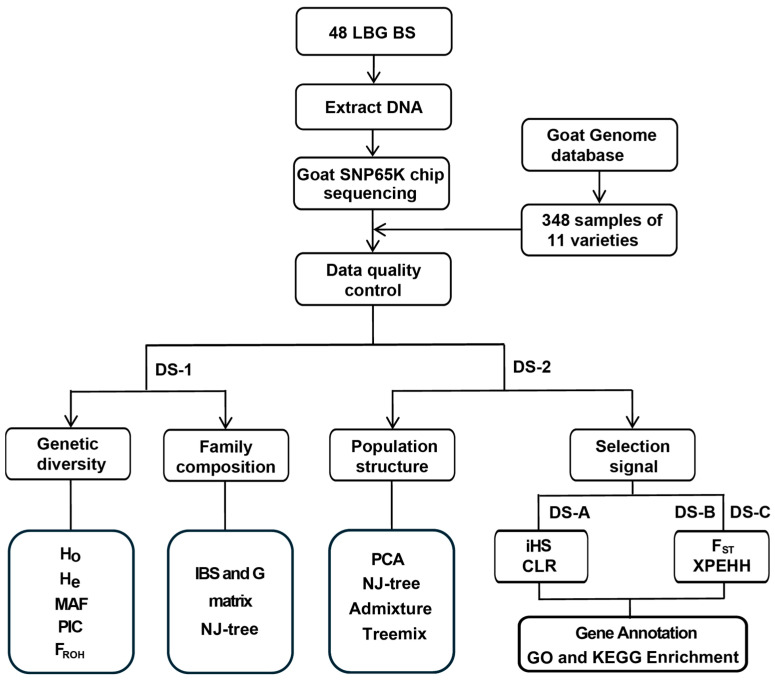
Workflow chart for this study.

**Figure 2 animals-14-03154-f002:**
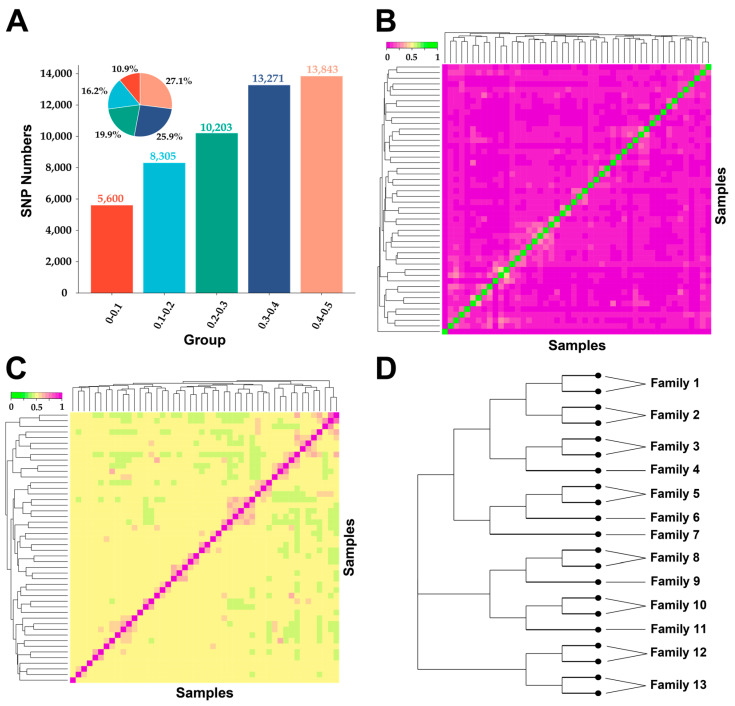
Genetic diversity and family composition of LBG population. (**A**) MAF interval distribution of LBG population; (**B**) 48 LBGs’ identity by state matrix; (**C**) 48 LBGs’ genetic relationship matrix; (**D**) NJ tree and division of families of male goats in LBG population.

**Figure 3 animals-14-03154-f003:**
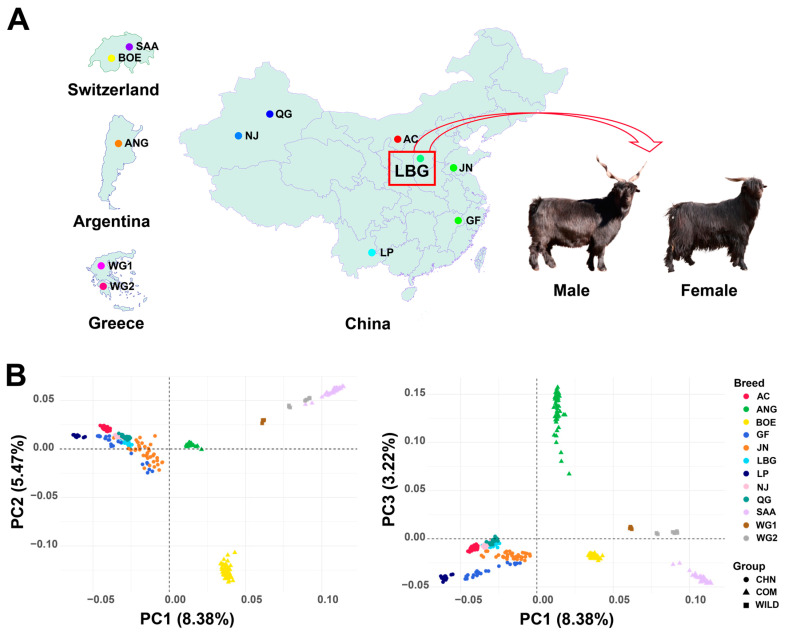
Geographical distribution and principal components of 12 goat populations. (**A**) Geographical regions of the country where the goat populations are located. (**B**) Principal component analysis of the goat populations. PC1 and PC2 on the left, PC1 and PC3 on the right. The 12 populations are represented by different colors, and the groups are represented by icons of different shapes. The circle represents Chinese (CHN) goats, the triangle represents commercial (COM) goats, and the rectangle represents wild (WILD) goats.

**Figure 4 animals-14-03154-f004:**
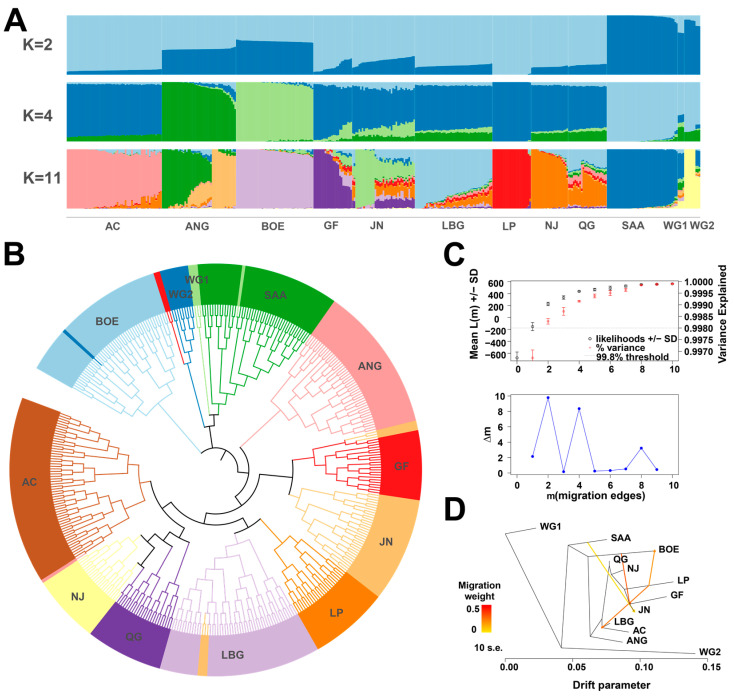
Population structure analysis of LBG and the remaining 11 goat breeds. (**A**) Ancestor component analysis of 12 goat populations. (**B**) NJ tree of 396 individuals in 12 goat populations. (**C**) Determination of the best migration model (m = 4 is the best migration model for 12 goat populations). (**D**) Migration model of 12 goat populations (m = 4).

**Figure 5 animals-14-03154-f005:**
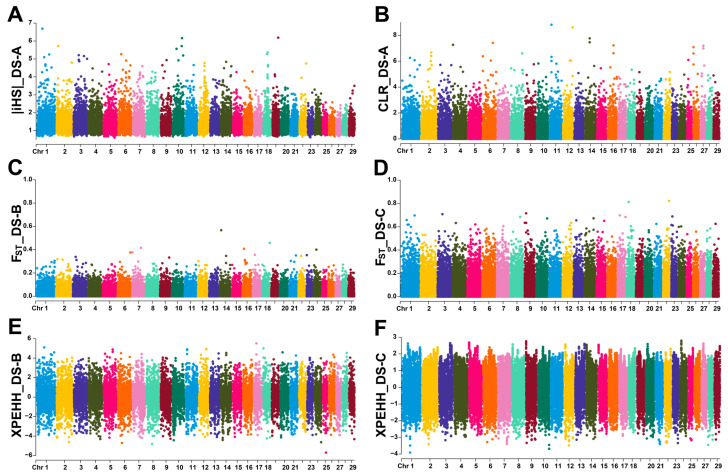
Genome-wide candidate region detection in LBG populations. (**A**) Whole-genome distribution of iHS scores in DS-A. (**B**) Whole-genome distribution of CLR values in DS-A. (**C**,**D**) Whole-genome distribution of F_ST_ values in DS-B and DS-C. (**E**,**F**) Whole-genome distribution of XP-EHH scores in DS-B and DS-C. Ten alternating colors are used to distinguish adjacent chromosomes.

**Figure 6 animals-14-03154-f006:**
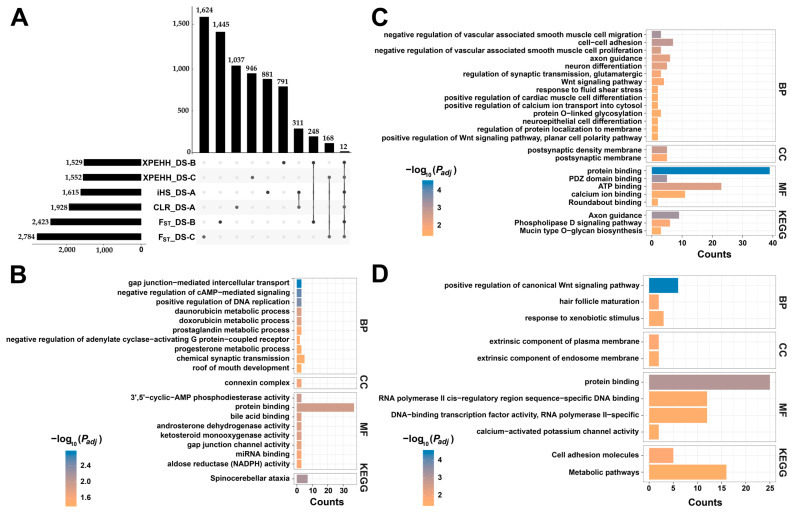
Gene annotation of the selected regions and functional enrichment analysis of the annotated genes. (**A**) Intersection of 6 candidate gene sets. (**B**) Functional enrichment results of candidate genes in DS-A. (**C**) Functional enrichment results of candidate genes in DS-B. (**D**) Functional enrichment results of candidate genes in DS-C.

## Data Availability

All data generated or analyzed during this study are included in this article/additional files.

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
