# Peer review of "Genetic Diversity and Selection Signatures of Lvliang Black Goat Using Genome-Wide SNP Data"

_animals, 2024, doi:10.3390/ani14213154_

Round 1

Reviewer 1 Report

Comments and Suggestions for Authors

Manuscript titled "Genetic diversity and selection signatures of Lvliang black goat using genome-wide SNP data" by Cai et al.

The authors assessed genetic diversity and identified selection signatures in the genome of the Lvliang black goat to further ascertain their association with production traits differentiating this goat breed from a set of 6 Chinese goat breeds, 3 commercial goat breeds and two wild goat breeds.

Sample used is enough. The comparative analyses performed allowed to identify genomic regions putatively associated with unique traits of this breed.

However, I have some methodological concerns that prevent me to recommend the acceptation of this paper for publication in Animals in its present form:

There are many methods useful for the identification of selection sweeps (e.g. Alvarez et al., 2020, Sci. Rep., 10, 2824, DOI:10.1038/s41598-020-59839-x). The authors applied four methods to identify selection signatures (L452-469). It would be necessary to explain the reason for choosing these four methods instead of others. Furthermore, the identification of selection sweeps implies multiple comparisons and, therefore, statistical methods for annotation cluster (e.g. that implemented in DAVID) of genes identified should be applied to ensure that their identification was not due to chance.

Minor concerns

L67. Colli et al.

L79. Although geographic isolation affects genetic diversity analyses, the development of the SNP array in Cosmopolitan breeds could be another problem.

L86. Delete “.”

L92-L96. Maybe it’s better to add references focusing on local goat breeds instead of those found in cattle.

L105-L107. Please, describe something about the breed itself. If it’s under a breeding or conservation programme, if the genetic background is well documented or not.

L125. ¿How many SNPs do you have?

L127. ¿How many SNPs include these SNP array? Please, describe how many SNPs were shared between both datasets.

L136. Please, clarify if you applied these thresholds to both datasets. I do not fully agree with filtering on MAF. Compared with classical editing strategies using Minor Allele Frequency and deviation from Hardy–Weinberg

proportions, the strategy to filter for Mendelian Errors (Arias, K.D., Álvarez, I., Gutiérrez, J.P., Fernández, I., Menéndez, J., Menéndez-Arias, N.A., Goyache, F. (2022) Understanding the occurrence of Mendelian Errors in SNP arrays data using a pedigree of Gochu Asturcelta pig: genomic alterations, family size and calling errors. Scientific Reports, 12:19686. DOI: 10.1038/s41598-022-24340-0) has been proven to be useful in removing genotyping errors while keeping putatively informative loci and datasets size.

L150. Which parameters do you use to assess ROH?

L177. Please, clarify it. I understand that the data is first imputed to avoid missing genotypes and then this dataset is used for all selection signature statistics, is that correct?

L186. Why you don’t overlap both statistics to find common SNPs?

L196. Is 50kb the medium distance between SNPs of your SNP array?

L198. Did you determine enriched functional terms?

L210. Please, clarify in the methodology section how many SNPs do you have before quality control.

L232. This number of SNPs is different than those describe in L210.

L244. I think the lowest cross-validation error was at K=8 (Figure 4A). Please plot only informative clusters. Maybe it’s also better to see if the differences between cross-validation errors were negligible or not.

L258-L260. Please, rewrite it.

L274. They are flanking regions of the SNPs, not candidate genomic regions itself.

L283. Of these 2069 candidate regions, how many overlap with those identified with iHS?. Maybe you could add a paragraph about this, and the genes found on these overlapping SNPs under selection.

L286-L289. Please, describe on which chromosome are identified each candidate gene.

L296. ”were located within this candidate region”

L338-L341. Have they formed a functional cluster?

L387. “so it could be that…”

L460-L464. And also because the SNP array is developed in Cosmopolitan breeds and it’s not specifical for this local breed.

Author Response

Response to Reviewer 1 Comments

Point 1: There are many methods useful for the identification of selection sweeps (e.g. Alvarez et al., 2020, Sci. Rep., 10, 2824, DOI:10.1038/s41598-020-59839-x). The authors applied four methods to identify selection signatures (L452-469). It would be necessary to explain the reason for choosing these four methods instead of others.

Response 1: Dear reviewer, regarding the formulation of the selection signal detection strategy you mentioned, we have some thought when designing the experiment. At present, the selection signal detection methods can be divided into three categories based on the detection principle, namely, based on allele frequency (CLR), based on linkage disequilibrium (XP-EHH) and based on population differentiation (FST). The three signal detection methods of CLR, XP-EHH and FST can be used to comprehensively detect population selection signals from multiple angles to achieve complementary effects. At the same time, we took into account that there will be certain differences in the results of selection signal analysis within and between populations, and the iHS method has a better detection effect on recent high-frequency variations within the population. Therefore, we chose these four selection signal methods to improve the credibility of the test results.

The use of several of these methods has also occurred in studies with goats.

Yao et al. used methods such as CLR, FST, and XP-EHH.(Yao et al. BMC Genomics. 2023 Mar 15;24(1):116. doi: 10.1186/s12864-023-09204-9. )

The iHS approach was used by Bertolini et al.(Bertolini et al. Genet Sel Evol. 2018. doi: 10.1186/s12711-018-0421-y.)

Taheri et al used iHS and XP-EHH methods.(Taheri et al. Anim Sci J. 2023. doi: 10.1111/asj.13864. PMID: 37560768.)

Point 2: Furthermore, the identification of selection sweeps implies multiple comparisons and, therefore, statistical methods for annotation cluster (e.g. that implemented in DAVID) of genes identified should be applied to ensure that their identification was not due to chance.

Response 2: Dear reviewer, I completely agree with your point of view and have followed your suggestions.

We supplemented the Fisher Exact test in our study and used the Fisher Exact test P value of 0.05 as the threshold for defining significantly enriched functional terms.

We then revised Figure 6B-6D and Table S10-12 in the article and made corrections in Materials and Methods 2.8.

Corrected to: "and Fisher exact test P value of 0.05 was used as the threshold for defining significantly enriched functional terms."

Point 3: L67. Colli et al.
Response 3: Dear reviewer, thank you for your correction.

We deleted ‘Colli et al.’ and changed the statement to ‘Goat SNP50 Bead Chip was used to evaluate...’.

Point 4: L79. Although geographic isolation affects genetic diversity analyses, the development of the SNP array in Cosmopolitan breeds could be another problem.

Response 4: Dear reviewer, thank you for your valuable suggestions.

The "development of the SNP array in Cosmopolitan breeds" you mentioned is an important factor that affects the genetic diversity results in the study. We apologize that our study did not take this aspect into consideration. We have supplemented this consideration in the limitations section of the discussion.

The added content is: "In addition, the SNP chip used in this study was developed based on Cosmopolitan breeds and is not specific to the LBG breed."

At the same time, we also found that it was inappropriate to cite the literature about pigs here, so we rewrote this paragraph.

Correction to: "A study using SNP chip technology found that most local populations in South Africa have gene exchange with commercial goat populations and have a common genomic ancestor, which is related to the preference of people in the region to breed goat breeds with good production efficiency. This study reminds policymakers to pay attention to the impact of hybridization and emphasizes the importance of protecting the core population of goats [6].”

Point 5: L86. Delete “.”

Response 5: Dear reviewer, thank you for your correction. We followed your suggestion and removed the extra "."

Point 6: L92-L96. Maybe it’s better to add references focusing on local goat breeds instead of those found in cattle.

Response 6: Dear reviewer, I completely agree with your point of view and have followed your suggestions.

We rewrote the paragraph using references about goats.

Corrected to: "These signals can effectively reveal the germplasm characteristics of a population when the population size is limited or the phenotypic data is incomplete. In the analysis of selection signals of 30 BoHuai goats, key genes related to lipid metabolism and disease resistance were successfully identified [8]. Similarly, analysis of the Pakistani Teddy goat population also identified multiple candidate genes related to important production traits [9].”

Point 7: L105-L107. Please, describe something about the breed itself. If it’s under a breeding or conservation programme, if the genetic background is well documented or not.
Response 7: Dear reviewer, thank you for your valuable suggestions.  

We apologize that due to our mistake, LBG was only described in the first paragraph of the introduction and this part was neglected.

There are currently few records of the genetic background of this breed, and the purpose of our research is to increase our understanding of the breed.

Thank you for your suggestion. We have added the information about LBG genetic background in the article.

The added content is:

1.”In 1979, the number of LBGs in Shanxi Province accounted for about 40% of the total number of goats in the Province. However, due to the extensive use of LBG in breeding with other breeds during the development of new varieties, the number of purebred LBGs has dropped sharply. In a 2009 survey of purebred LBG, only about 420 were found. The LBG was once on the verge of extinction. Therefore, LBG, an important local goat breed resource, has received attention from the country, and an LBG conservation farm was established in 2010 to carry out conservation work.”

2.”However, as an important genetic resource of Chinese goats, the genetic mechanism and germplasm characteristics of LBG have not been fully elucidated, which limits the effectiveness of conservation and breeding work.”

Point 8: L125. How many SNPs do you have?

Response 8: Dear reviewer, we are deeply sorry that due to our mistake we did not include this information in the article.

This study used a 65K SNP chip to detect 59,727 SNP sites from the LBG population, and we have supplemented the relevant content in the article.

The added content is: ”and finally 59727 SNP sites were detected (DS-1).”

Point 9: L127. How many SNPs include these SNP array? Please, describe how many SNPs were shared between both datasets.

Response 9: Dear reviewer, we are deeply sorry that due to our mistake we did not include this information in the article.

We detected 59,727 SNPs in the LBG population, and after merging the LBG population with other goat population datasets, we obtained 43,171 shared SNPs.

We have added the corresponding content to the article. The added content is:

1.”and finally 59727 SNP sites were detected (DS-1).”

2.”We extracted the SNP sites common on the two chips and obtained a total of 43,171 SNP sites (DS-2) .”

Point 10: L136. Please, clarify if you applied these thresholds to both datasets. I do not fully agree with filtering on MAF. Compared with classical editing strategies using Minor Allele Frequency and deviation from Hardy–Weinberg proportions, the strategy to filter for Mendelian Errors (Arias, K.D., Álvarez, I., Gutiérrez, J.P., Fernández, I., Menéndez, J., Menéndez-Arias, N.A., Goyache, F. (2022) Understanding the occurrence of Mendelian Errors in SNP arrays data using a pedigree of Gochu Asturcelta pig: genomic alterations, family size and calling errors. Scientific Reports, 12:19686. DOI: 10.1038/s41598-022-24340-0) has been proven to be useful in removing genotyping errors while keeping putatively informative loci and datasets size.

Response 10: Dear reviewer, thank you for your suggestion. 

First, we need to clarify the filtering strategy used in this study. We used two datasets in this study, a single LBG population dataset DS-1 and a multi-population dataset DS-2. Four filtering criteria, such as MAF, were initially applied to both datasets. Then, before specific analysis, we implemented a strategy for filtering sites that deviated from Hardy-Weinberg equilibrium for DS-1 and a strategy for filtering high LD sites for DS-2. We did this for two reasons:

1.The strategy of filtering sites that deviate from Hardy-Weinberg equilibrium (HWE) is not applicable to data sets of multiple populations for two reasons. First, different populations may have different genetic backgrounds and allele frequencies. Sites that appear to deviate from HWE in one population may be normal in another population. Second, different populations may have genetic heterogeneity, that is, the same gene locus may behave differently in different populations. Strict HWE filtering may mistakenly remove variants that are meaningful in a specific population. Therefore, we did not adopt the strategy of filtering HWE in DS-2.

  1. High LD usually means that there is a strong genetic association between sites, which may reflect the same genetic information. Therefore, if these sites are included in the analysis, many repeated tests will be performed, which may lead to an increase in false positive results. By selecting sites that are highly representative and independent of each other, the population structure can be estimated more accurately. Therefore, we adopt the strategy of filtering high LD sites in DS-2.

We are sorry that our incorrect statement has caused you trouble. We have corrected the article to make it clearer.

Corrected to: "We extracted the SNP sites common on the two chips and obtained a total of 43,171 SNP sites (DS-2) . We used PLINK v1.9 software [12] to perform quality control on the DS-1 and DS -2 datasets.”

Secondly, we apologize for our mistake in not adopting the strategy of filtering Mendelian errors in our study, which is essential for family-related analysis. Therefore, we performed relevant filtering on the DS-1 original data set, but found no sites and samples that were removed. We also followed your advice and added the content of filtering Mendel's errors into the article.

The added content is: ”Specifically, we further used the parameters '--hwe 0.000001' and '--me 0.05 0.1' to exclude sites that deviated from Hardy-Weinberg equilibrium and Mendelian error in DS-1 before conducting the analysis.”

Finally, regarding the case where MAF and HWE filtering are not recommended as mentioned in the literature (DOI: 10.1038/s41598-022-24340-0), after careful consideration, we believe that it is feasible to retain MAF and HWE filtering in this study. The reason is that the filtering criteria of MAF (p value ≤ 0.05) and HWE (p value ≤ 0.001 or ≤ 0.0001) used in the above literature are very strict, which is why a large number of SNPs were filtered in the literature. In this study, our standards are not strict, and the thresholds are MAF (p value ≤ 0.01) and HWE (p value ≤ 0.000001). These two filtering criteria removed 3695 and 289 SNP sites for DS-1, respectively, and had little impact on the data set. Therefore, we believe that this is an acceptable and reasonable quality control standard.

Point 11: L150. Which parameters do you use to assess ROH?

Response 11: Dear reviewer, we apologize that we did not evaluate the ROH segment in our study and only calculated the inbreeding coefficient of the LBG population based on ROH.

Point 12: L177. Please, clarify it. I understand that the data is first imputed to avoid missing genotypes and then this dataset is used for all selection signature statistics, is that correct?

Response 12: Dear reviewer, We apologize for the confusion caused by our incorrect statement. We must clarify one point: In this study, we only performed genotype filling in the iHS and XP-EHH methods, because both methods rely on the accurate identification of long haplotypes. If there are too many missing values ​​in the genotype data, the continuity of the haplotype may be interrupted, resulting in a decrease in statistical power. Through genotype filling, these haplotypes can be more accurately identified and analyzed, thereby improving the power of statistical tests. In addition, in XP-EHH analysis, it is often necessary to compare the haplotype structure between different groups. Genotype filling can ensure that the two comparison groups have similar data quality, making the comparison more fair and accurate.

Beagle software is a genotype filling tool that can predict missing or incorrect genotypes based on the genotype data of other individuals through a probability model, thereby improving the quality of the entire data set. Therefore, in this study, we used Beagle software for gene filling.

In order to make the article more clear, we have corrected the article.

Corrected to: ”iHS first used Beagle v5.2 software [23] to fill in missing genotypes and infer the haplotypes of each chromosome.”

Corrected to: ”and XPEHH first needed to use Beagle v5.2 software to fill in the genotype data of the target population and the reference population and infer the haplotype”

Point 13: L186. Why you don’t overlap both statistics to find common SNPs?

Response 13: Dear reviewer, we apologize for the unclear content of the article due to our incorrect description. In the results 3.4 of the article, we annotated the regions where the SNPs detected by different analysis methods were located, and then overlapped the resulting genes. Specific content of the article:

1.”We used iHS and CLR methods to jointly identify 310 candidate genes(Table S9) in DS-A.”

2.”DS-B identified 248 identical candidate genes under the FST and XPEHH methods,”

3.“Among the candidate genes identified by DS-C, 168 genes were shared,”

Point 14: L196. Is 50kb the medium distance between SNPs of your SNP array?

Response 14: Dear reviewer, thank you for your comments.

In the SNP data used in our study, the average distance between SNP sites is about 41kb. Therefore, we defined the region 50kb upstream and downstream of the SNP site as the candidate region.

Point 15: L198. Did you determine enriched functional terms?

Response 15: Dear reviewer, we apologize that we did not write about enrichment analysis here, but wrote this part in Results 3.4. This study conducted functional enrichment analysis on overlapping genes annotated by multiple methods. The specific content of the article is:

1.”The results (Table S10) showed that the 311 candidate genes were mainly enriched in 20 significant items...”

2.”Function enrichment analysis showed that 248 genes were mainly enriched in 24 significant items (P < 0.05), including...”

3.”These candidate genes were mainly enriched in...”

Point 16: L210. Please, clarify in the methodology section how many SNPs do you have before quality control.

Response 16: Dear reviewer, we apologize for the unclear content of the article due to our incorrect description. We have changed the wording of the article to make it clearer. The changes are:

1.”and finally 59727 SNP sites were detected (DS-1).”

2.”We extracted the SNP sites common on the two chips and obtained a total of 43,171 SNP sites (DS-2) “

Point 17: L232. This number of SNPs is different than those describe in L210.
Response 17: Dear reviewer, we apologize for causing you trouble due to our incorrect statement. We would like to clarify here: In the study, the dataset (DS-1) we used in Result 3.1 is the SNP data of a single LBG population, which is used to analyze the genetic diversity and pedigree composition of LBG population; in Result 3.2, we used a combined dataset of 12 goat populations (DS-2) to investigate the genetic clustering pattern of LBG and 11 goat breeds, so the number of SNPs in the two places will be different. To make the statement clearer, we have changed the article.

Corrected to:

1.”After quality control of DS-1, 51,222 SNPs were retained.”

2.”After quality control of DS-2, 37,882 SNPs were obtained for subsequent analysis.”

Point 18: L244. I think the lowest cross-validation error was at K=8 (Figure 4A). Please plot only informative clusters. Maybe it’s also better to see if the differences between cross-validation errors were negligible or not.
Response 18: Dear reviewer, thank you for your correction. We misjudged the lowest cross-validation error value, and K=8 is a more reasonable choice. After your correction, we also realized that the consideration of the difference between cross-validation errors is not the main information we are concerned about and is not very helpful for the research results. Therefore, we followed your advice and removed the analysis of cross-validation errors and corrected the relevant content of the article.

Corrected to: ”Then, ancestral component analysis (K=2~20) was performed using ADMIXTURE v1.30 [19] to verify the clustering patterns among populations and infer the possible sources of genetic variation in the test populations.”

Finally, we re-draw Figure 4 according to your suggestion, retaining only the content when K is 2, 4, and 11.

Point 19: L258-L260. Please, rewrite it.

Response 19: Dear reviewer, I completely agree with your point of view and have followed your suggestions. Rewrite the content as follows:

”ADMIXTURE software was further used to quantify the degree of admixture between populations. When K was 2, it was observed that the seven Chinese goat breeds were clearly distinguished from the other five foreign goat breeds, but except for the LP group, the Chinese goat breeds were mixed with a small amount of genetic background of foreign goat breeds; when K was 4, the LBG, NJ and QG groups showed similar mixing component ratios, and only when K was 11 did LBG show a different mixing component ratio from the other groups (Figure 4A).”

Point 20: L274. They are flanking regions of the SNPs, not candidate genomic regions itself.

Response 19: Dear reviewer, thank you for your correction. After careful consideration, we believe that the term ‘candidate region’ can make the article clearer. Therefore, we added the content defining the flanking regions of SNPs as candidate regions in Materials and Methods. The added content is:

1.”The windows with the top 5% |iHS| values were defined as candidate regions”

2.”and the windows with the top 5% CLR values were defined as candidate regions.”

3.”After sorting, the 50 kb genomic intervals upstream and downstream of the top 5% SNP sites were identified as candidate regions.”

Point 21: L283. Of these 2069 candidate regions, how many overlap with those identified with iHS?. Maybe you could add a paragraph about this, and the genes found on these overlapping SNPs under selection.
Response 21: Dear reviewer, we apologize for the unclear content of the article due to our incorrect description. We need to clarify that the 2069 candidate regions in the article refer to the 2069 candidate regions detected by the FST method in both DS-A and DS-B datasets, while the XPEHH method detected 1805 candidate regions and 1938 candidate regions in DS-A and DS-B datasets, respectively. Regarding the description of overlapping regions, we also realize that this is easy to cause misunderstanding. We have supplemented it in Materials and Methods 2.7 in the article, highlighting which datasets the four methods analyzed respectively, making the content easier to understand. Thank you for your correction. Corrected to:

1.”To detect the selection signal in the LBG population, we constructed three datasets (DS): DS-A: LBG population; DS-B: LBG and six Chinese goat populations; DS-C: LBG and three commercial goat populations.”

2.”To evaluate the characteristics of selection between LBG and different populations, we used two complementary methods, namely the index of genetic differentiation between populations (FST) and selection scanning analysis based on population haplotypes (XP-EHH), to analyze the selection signals of DS-B and DS-C.“

Point 22: L286-L289. Please, describe on which chromosome are identified each candidate gene.
Response 22: Dear reviewer, we have corrected the article according to your comments. Corrected to:

1.”of which the RPTOR gene was annotated on this candidate region of chromosome 19 and the MYO9A gene was annotated on this candidate region of chromosome 10.”

2.”of which the BIRC6 gene was annotated on this candidate region of chromosome 11 and the DIAPH3 gene was annotated on this candidate region of chromosome 12.”

3.”and only the NEDD4 gene was located in this candidate region of chromosome 10.”

4.”among which the LOC106502907 gene was annotated in this candidate region of chromosome 2, and the FAM47E, SCARB2, and LOC102177148 genes were annotated in this candidate region of chromosome 6.”

5.”The LSAMP gene was annotated in this region of chromosome 1, the TMEM132D gene was annotated in this region of chromosome 17, and the LOC102180550 and LOC102180813 genes were annotated in this candidate region of chromosome 18.”

Point 23: L296. ”were located within this candidate region”
Response 23: Dear reviewer, thank you for your correction.

Corrected to: ”and no genes were located within this candidate region.”

Point 24: L338-L341. Have they formed a functional cluster?
Response 24: Dear reviewer, thank you for your comments. In this part, we performed functional enrichment analysis on the overlapping genes detected by different selection signal methods and obtained multiple functional entries, such as “Roof of Mouth Development”, “Bile Acid Binding”, etc.

We grouped genes with similar effects or enriched in the same pathway. After our sorting and screening, we found that genes such as CFL2 and INSIG1 were related to growth traits, and other genes also showed similar performance.

Point 25: L387. “so it could be that…”
Response 25: Dear reviewer, thank you for your correction.

Corrected to: ”the LBG population so it could be that a special population...”

Point 26: L460-L464. And also because the SNP array is developed in Cosmopolitan breeds and it’s not specifical for this local breed.
Response 26: Dear reviewer, thank you for your valuable suggestions. Your suggestions have greatly helped the rigor of the article, and we have supplemented the content of the article based on your opinions.

The added content is: “In addition, the SNP chip used in this study was developed based on Cosmopolitan breeds and is not specific to the LBG breed.”

Reviewer 2 Report

Comments and Suggestions for Authors

In this paper, the genetic situation of the LGB breed, a local goat population in China, is characterised using genomic data. Additionally, a comparison is made with other genetically related populations. Finally, a search for selection signals is carried out using the information from existing samples in a selection of these breeds.

Although the methodology for analysing the genomic data is adequate and very broad, and the workflow is very clear, some small extra genomic analyses should be added, especially the detection of possible population bottlenecks and the estimation of the effective size in this population (or in the DS-1 populations in order to also establish a comparative analysis).

In contrast, the estimation of the PIC, like the comparison with the SSR, does not contribute anything to the objective of the work, and should be eliminated.

Although there is an introduction on the breed, it should be expanded: what is the population census? What is the census evolution in recent decades? Has a census bottleneck been detected? This data could be compared with the analysis obtained with the genomic data.

Given its genetic proximity to other Chinese populations, can it be indicated that it comes from the same population and that it has been differentiated by drift (assuming that there is no selection plan with differential objectives)?.

How were the animals of the LBG breed sampled? Given that the number of samples analyzed (49) is very limited, the sampling carried out is essential to collect all the variability of the breed, and to avoid the presence of genetically related individuals… , from how many flocks were these samples obtained?… from the ibs analysis it can be deduced the absence of close relationships, is this fact a normal situation of the breed despite its endangered status or is it due to the breeding farm practices that the authors recognize? Or is it the result of sampling to eliminate related animals? Based on this answer, the statement that there are no problems of inbreeding depression in the 364 line should or should not be eliminated.

Based on genomic data, an estimate of the kinship relationships between animals is made. It is understood that this is done because there is no reliable pedigree. Is there any kind of control of the filiation of the animals? If there is no genealogical control, the expression pedigree should not be used, since this term assumes kinship relationships based on genealogical records that are understood not to exist, or at least not exist with any kind of mechanism to ensure their reliability.

The authors use the –cv flag of the admixture package, which allows cross-validation of the results. How does this option allow ancestral component analysis?

For the Identification and functional annotation of candidate genes, the selected genomic intervals were mapped to the goat reference genome Genome assembly ARS1.0, when ARS1.2 is available, which updates ARS1.0, can you justify why it has not been used?

L368-369 The statement “We divided the LBG breeding population into 13 independent pedigrees based on the standard of inter-individual kinship coefficient greater than 0.1, which exceeds the minimum standard of at least 6 different pedigrees required by national breeding sites” should be explained, since the international criteria for establishing the risk level of a population do not include anything similar. Among these, one of the most important is the effective number or size. It would be very interesting to estimate it from genomic information.

From the point of view of the objectives of this work, there are much more important results than the division into 13 pedigrees (in addition, this way of expressing this information is questionable). It should be eliminated from the conclusions, and replaced by other parameters such as FROH (as well as Ne and the presence or absence of bottlenecks) if it is to indicate that the variability that the population still retains is still high and that therefore conservation-improvement strategies can be established in this population.

Author Response

Response to Reviewer 2 Comments

Point 1: Although the methodology for analysing the genomic data is adequate and very broad, and the workflow is very clear, some small extra genomic analyses should be added, especially the detection of possible population bottlenecks and the estimation of the effective size in this population (or in the DS-1 populations in order to also establish a comparative analysis). 

Response 1: Dear reviewer, thank you for your valuable suggestions.

Following your comments, we have supplemented the analysis on effective population size in the study and added relevant content to the article.The added content is:
1. Located in Method 2.4 ”Afterwards, we used PLINK v1.9 software to perform linkage disequilibrium screening on the SNP data with the parameter '-indep-pairwise 50 10 0.2'. For the 26,913 SNPs remaining after screening, we used SNeP v1.1 software [13] to estimate the effective population size (Ne) of the historical generations of the LBG population, and used the linkage disequilibrium method of NeEstimator v2.1 software [14] to estimate Ne of the current generation [15].”

  1. Located in Result 3.1 “The Neof the LBG conservation population in 2024 was estimated to be 119 based on the linkage disequilibrium method. It was also found that the effective population size gradually decreased with the decrease of generations (Figure S1). The effective population sizes of the LBG populations 13 and 98 generations ago were 192 and 1161, respectively.”
  2. Located in the first paragraph of Discussion ”In addition, the Neof the LBG population in this study showed a downward trend with the decrease in the number of generations, and it may have experienced a population bottleneck in the past, which is consistent with the actual situation.”

Point 2: In contrast, the estimation of the PIC, like the comparison with the SSR, does not contribute anything to the objective of the work, and should be eliminated.

Response 2: Dear reviewer, thank you for your correction. We followed your advice and moved the PIC-related analyses and results out of the article.

Point 3: Although there is an introduction on the breed, it should be expanded: what is the population census? What is the census evolution in recent decades? Has a census bottleneck been detected? This data could be compared with the analysis obtained with the genomic data.
Response 3: Dear reviewer, thank you for your valuable suggestions. We do know from historical documents that there is a census bottleneck for Luliang Black Goats. Therefore, we have made corresponding additions in the first paragraph of the introduction of the article. The added content is:

“In 1979, the number of LBGs in Shanxi Province accounted for about 40% of the total number of goats in the province. However, due to the extensive use of LBG in breeding with other breeds during the development of new varieties, the number of purebred LBGs has dropped sharply. In a 2009 survey of purebred LBG, only about 440 were found. The LBG was once on the verge of extinction. Therefore, LBG, an important local goat breed resource, has received attention from the country, and an LBG conservation farm was established in 2010 to carry out conservation work.”

Point 4: Given its genetic proximity to other Chinese populations, can it be indicated that it comes from the same population and that it has been differentiated by drift (assuming that there is no selection plan with differential objectives)?.

Response 4: Dear reviewer, thank you for your comments.

Regarding "it comes from the same population", in the population structure analysis of 12 goat populations, we speculated based on the results that LBG and NJ and QG populations may have a common ancestor, and there is gene exchange between LBG and LP populations. However, there is no definite evidence in this study that LBG and the other three Chinese breeds come from the same population.

Point 5: How were the animals of the LBG breed sampled? Given that the number of samples analyzed (49) is very limited, the sampling carried out is essential to collect all the variability of the breed, and to avoid the presence of genetically related individuals… , from how many flocks were these samples obtained?… from the ibs analysis it can be deduced the absence of close relationships, is this fact a normal situation of the breed despite its endangered status or is it due to the breeding farm practices that the authors recognize? Or is it the result of sampling to eliminate related animals? Based on this answer, the statement that there are no problems of inbreeding depression in the 364 line should or should not be eliminated.

Response 5: Dear reviewer, we apologize for not describing the sample selection process in detail in our study. We will clarify this matter:

One of the purposes of this study is to evaluate the conservation effect of the Luliang Black Goat population, so the samples used are from the core conservation population of the conservation farm. In the selection of male goats, we selected all the breeding male goats, 21 in total, in order to cover all families; in the selection of female goats, we randomly sampled the core group of 211 female goats to reduce the possible impact of sample selection on the results. We have made a correction in the article on this point.

Corrected to: “We selected all male goats (21) and some female goats (27) from the LBG core conservation population, a total of 48 LBGs for venous blood sample collection.”

We believe that the conclusion of low inbreeding degree in the IBS analysis is caused by the practice of the breeding farm. In the breeding farm, each Luliang Black Goat has a fixed number and pedigree information, and each mating is also recorded. The breeder will use this information to guide mating. This may be the reason for the low inbreeding degree of the group in the research results. Thank you for your correction on the statement that "there is no problem of inbreeding depression". We have realized that the low kinship between individuals does not mean that there is no inbreeding depression in the group. We have corrected the article according to your opinion.

Corrected to: “This low inbreeding degree and kinship of the LBG population are conducive to maintaining the long-term survival and reproduction of the LBG population [31].”

Point 6: Based on genomic data, an estimate of the kinship relationships between animals is made. It is understood that this is done because there is no reliable pedigree. Is there any kind of control of the filiation of the animals? If there is no genealogical control, the expression pedigree should not be used, since this term assumes kinship relationships based on genealogical records that are understood not to exist, or at least not exist with any kind of mechanism to ensure their reliability.

Response 6: Dear reviewer, thank you for your valuable suggestions. We apologize for causing you trouble due to our incorrect description.We did not perform any form of control for animal kinship in the sample selection process. We are aware of this error and have replaced "pedigree" with "family" throughout the article.

Point 7: The authors use the –cv flag of the admixture package, which allows cross-validation of the results. How does this option allow ancestral component analysis?
Response 7: Dear reviewer, we apologize for causing you trouble due to our incorrect description. The detailed command for our ancestral component analysis is:

for K in {2..20}

do

   admixture --cv input.bed $K | tee log${K}.out

done

We also realize that the consideration of the differences between cross-validation errors is not the main information of this study. Therefore, we have removed the analysis of cross-validation errors and corrected the relevant content of the article and Figure 4.

Corrected to: ”Then, ancestral component analysis (K=2~20) was performed using ADMIXTURE v1.30 [19] to verify the clustering patterns among populations and infer the possible sources of genetic variation in the test populations.”

Point 8: For the Identification and functional annotation of candidate genes, the selected genomic intervals were mapped to the goat reference genome Genome assembly ARS1.0, when ARS1.2 is available, which updates ARS1.0, can you justify why it has not been used?

Response 8: Dear reviewer, we are sorry that since the 65K SNP chip data and downloaded SNP data used in this study are aligned with the ARS1.0 reference genome, we can only use the corresponding reference genome for gene annotation.

Point 9: L368-369 The statement “We divided the LBG breeding population into 13 independent pedigrees based on the standard of inter-individual kinship coefficient greater than 0.1, which exceeds the minimum standard of at least 6 different pedigrees required by national breeding sites” should be explained, since the international criteria for establishing the risk level of a population do not include anything similar.

Response 9: Dear reviewer, thank you for your valuable suggestions. The minimum standard in the article comes from China's "Regulations on the Management of Livestock and Poultry Genetic Resources Conservation Farms, Protection Areas and Gene Banks". Based on your suggestions, we realized that this standard does not meet international standards, so we have corrected the content of the article.

Corrected to: “We divided the LBG breeding population into 13 independent family lines based on the clustering results of male goats and the relationship between individuals to provide a reference for subsequent breeding work.”

In addition, we have supplemented the article with information about the effective population size. For more information, please see Response 1.

Point 10: Among these, one of the most important is the effective number or size. It would be very interesting to estimate it from genomic information. From the point of view of the objectives of this work, there are much more important results than the division into 13 pedigrees (in addition, this way of expressing this information is questionable). It should be eliminated from the conclusions, and replaced by other parameters such as FROH (as well as Ne and the presence or absence of bottlenecks) if it is to indicate that the variability that the population still retains is still high and that therefore conservation-improvement strategies can be established in this population.

Response 10: Dear reviewer, thank you for your valuable suggestions. We realize that the work of dividing the family does not reflect the high variability of the LBG population. We have corrected the content of the article according to your comments.

Corrected to: “We divided the LBG breeding population into 13 independent family lines based on the clustering results of male goats and the relationship between individuals to provide a reference for subsequent breeding work.”

In addition, we have supplemented the article with information about the effective population size. For more information, please see Response 1.

Reviewer 3 Report

Comments and Suggestions for Authors

The manuscript (MS) provides population genetic and genome-wide association analyses of  Lvliang black goat.
The authors also looked for the genes located in the top regions, found
by different algorithms and setups, and described these genes based on
their known functions.

Descriptions are well-detailed, they will be useful for the readers. Analyses
were performed cautiously, and their results are discussed from several
points of views.

I have noticed several things to remedy, but they can be considered as
minor modifications. I suggest the MS for publication after the proposed
modifications are completed.

Suggestions, notices:

Line 18: ‘seed conservation’ looks strange in an article dealing with animals, not plants. It’s usage is more abundant in plant breeding. Here I suggest to use only the word 'conservation' or the phrase ‘conservation of the core population’.

Line 20: How the population was divided? No explanation might be good in the Simple Summary, but it must be written in other section(s).

Lines 22-23: ‘Finally … analysis.’. Germplasm characteristics determine the genetic structure as well. The exploration of selection signals gives us a kind of characteristic. I suggest using the sentence 'We also looked for selection signals and performed functional enrichment analysis'. However, I have doubts that ‘functional enrichment’ are the proper words in Simple Summary. Can you find less technical words?

Line 24: Please delete ‘new strength’.

Line 29: Please delete ‘seed’.

Line 30: Delete  ‘conservation’. Conservation has no genetic diversity.

Line 32: Information is missing on how the population was divided into pedigrees. ‘
‘ It was found that the LBG population had high genetic diversity and distant relationships between 31 individuals, and 13 pedigrees were successfully divided.’ should be changed to ‘
‘Based on the high genetic diversity and LBG population had and distant relationships between 31 individuals  of LBG breed, they were classified into 13 pedigrees.’
Later on, exact description is necessary how this division of 31 animals into 13 pedigrees was achieved.

Line 33: Give the names of the breeds.

Line 38: Give the names of the commercial breeds.

Line 40: Instead of ‘ Finally, we functionally enriched these genes and…’
write ‘Finally, we performed functionally enrichment analysis on these genes and…’

Lines 40-41: Delete ‘In summary, the LBG population has achieved good results in seed conservation and exhibited unique genetic structure and selection characteristics.’. It gives no additional information.

Line 57: what does ‘disorderly introduction’ mean? Please elaborate.

Line 60: Please delete ‘seed’. Give the reference, where ‘the list of Chinese livestock and poultry breed resources’ can be accessed.

Line 65: Delete ‘means’.

Line 67: Delete ‘Colli used’, Write: ‘…Chip was used to evaluate…’.

Line 71: This part is unnecessary, please delete: ‘providing valuable genetic evidence for in-depth analysis of the domestication
process and migration routes of goats’

Lines 78-83: This MS is focusing on goat. The pig aspect should be removed. If it remains here, why did the authors stop here? Horse, sheep, and cattle references should also be given. That is why I say there is no need to give this example on pigs.

Lines 89-96: References on non-Chinese breeds should also be given.

Line 92: The reference 8 on cattle is raising the same issue, that was mentioned about pigs. This part could be deleted or give non-chinese examples as well.

Line 109: Replace 'conservation’ with ‘core’.

Line 112: Delete ‘dominant’.

Line 115: Change ‘germplasm characteristics’ to ‘on goat genome’.

Line 120: Delete ‘from’. Give the company name, and the country as well.

Line 122: Change ‘standard’ to ‘…(the accepted value to include the sample in the analyses was …)

Line 135: Please change ‘…to the two chip data were…’ to ‘on the two chips were…’

Line 148: ‘length’ and ‘fragment’: Use plural forms.
Lines 157-158: Give details, how pedigrees were determined.

Line 170: There is just one root in one analysis. Please specify, when WG1 and WG2 were used in which migration analyses.

Line 224: 13 Pedigrees are Based on which characteristic of the tree? Please mark the branches, which are thought to be representing the pedigrees, with thicker lines, As for the colours of the samples, it seems, they are not uniformly grouped on a given branch.

Line 227: IBS ranges from 0 to 1. On the Figure I see it is from -1 to 1. Please remedy.

Figure 3: It is not easy to distinguish similar colours (additional information: the writer of these rows went through a colour blindness test, without a mistake). Please introduce different forms of markers. E.g.: rectangular, diamond, triangle, etc.  Please avoid to have the same marker type for two similar colours.

Line 362: The value 0.024 is not lower than 0.017. Please rewrite, and describe what the 0.024 value indicates in the term of health and adaptibility comparing them to the other breeds.

Line 367: This description is missing from the Materials and methods. Please insert this explanation there as well, with more technical details. 

Line 369: ‘…by national breeding sites.’ Please insert a reference to the end of this sentence.

Line 375: Do the authors need the word 'relatively'? If yes, why is the distance called ‘relatively distant’? Examples from other research or from the presented data should be given.

Line 383: Please replace ‘my country’ with a less personal term.

Line 386: Again, the word 'relatively'. See the comment above given for Line 375.

Lines 389 and 391: Is ‘may’ the proper word? Isn’t it known, from historical data, that mixing occured? The results presented here supports more than ‘may’. Can you say LBG is a unique population based on the results? Does it support the uniqueness or not?

Line 438: Change ‘In addition, to fully understand the germplasm characteristics of LBG, we…’ to ‘In addition, to further explore the genome of LBG, we…’

Line 450: Delete ‘…Darwish's study pointed out that…’

Line 452: ‘What do 'common genes' mean here? Are they common based on what? Or are these the genes found by all the approaches presented in this study? Please clarify.

Line 460, first sentence: It is enough to write: ‘Limitations of this study are...’.

Line 463: Has the LBG whole genome sequence been published/reported somewhere?
If not, the 'sequencing' word seems to be more appropriate than ‘resequencing’.

Line 467: Replace ‘it’ by ‘breed’.

Line 468: Instead of ‘distantly related, and the seed conservation effect was good.’
write ‘distantly related.The conservation effort was effective.’

Lines 468-469: Delete ‘In addition, LBG has a unique genetic structure and selection characteristics.’.

Line 470: Instead of ‘identified selected genes’ it is more appriate to write ‘identified genes under selection’.

Line 471: Delete ‘superior’.

Lines 471, 472: Delete ‘study further analyzed the germplasm characteristics and genetic mechanisms of LBG,’.

Line 472: Insert a ‘to’ before ‘support’, delete ‘for’. Replace ‘…protection and breeding of LBG,’ with ‘...protection and improvement of LBG.

Comments on the Quality of English Language

The overall readability is good, but now and then a few sentences require some intervention to make them clearer or fluent.

Author Response

Response to Reviewer 3 Comments

Point 1: Line 18: ‘seed conservation’ looks strange in an article dealing with animals, not plants. It’s usage is more abundant in plant breeding. Here I suggest to use only the word 'conservation' or the phrase ‘conservation of the core population’.
Response 1: Dear reviewer, thank you for your correction. We followed your suggestion and replaced ‘seed conservation’ with ‘conservation’ in the article.

Point 2: Line 20: How the population was divided? No explanation might be good in the Simple Summary, but it must be written in other section(s).

Response 2: Dear reviewer, thank you for your valuable suggestions.  

We followed your suggestion and made additions to Abstract and Materials and Methods. The added content is:

1.”According to the clustering results of male goats and the relationship between individuals, the LBG population was divided into 13 families.”

2.”Finally, the LBG population was divided into families based on the clustering results of male goats and the criterion that individuals with a kinship greater than 0.1 belonged to the same family.”

Point 3: Lines 22-23: ‘Finally … analysis.’. Germplasm characteristics determine the genetic structure as well. The exploration of selection signals gives us a kind of characteristic. I suggest using the sentence 'We also looked for selection signals and performed functional enrichment analysis'. However, I have doubts that ‘functional enrichment’ are the proper words in Simple Summary. Can you find less technical words?
Response 3: Dear reviewer, We completely agree with your point of view and have followed your suggestions.

Corrected to: ”Finally, we also looked for selection signals in the LBG population and obtained multiple functional terms.”

Point 4: Line 24: Please delete ‘new strength’.

Response 4: Dear reviewer, thank you for your correction. We have followed your suggestion and removed the word ‘new strength’ from the article.

Point 5: Line 29: Please delete ‘seed’.

Response 5: Dear reviewer, thank you for your correction. We have followed your suggestion and removed the word ’seed’ from the article.

感谢您的指正。我们遵循您的建议在文章中删除了‘seed’一词。

Point 6: Line 30: Delete  ‘conservation’. Conservation has no genetic diversity.
Response 6: Dear reviewer, thank you for your correction. We have followed your suggestion and removed the word ‘conservation’ from the article.

Point 7: Line 32: Information is missing on how the population was divided into pedigrees. ‘ It was found that the LBG population had high genetic diversity and distant relationships between 31 individuals, and 13 pedigrees were successfully divided.’ should be changed to ‘Based on the high genetic diversity and LBG population had and distant relationships between 31 individuals  of LBG breed, they were classified into 13 pedigrees.’ Later on, exact description is necessary how this division of 31 animals into 13 pedigrees was achieved.
Response 7: Dear reviewer, we are deeply sorry for the inconvenience caused to you due to the lack of article information. This study divided families according to the clustering results of male goats and the standard that individuals with a kinship relationship greater than 0.1 were considered to be in the same family. We have added this information in Materials and Methods 2.5 and have corrected the article according to your suggestion.

Corrected to:

1.”It was found that the LBG population had a high genetic diversity and a low degree of inbreeding. According to the clustering results of male goats and the relationship between individuals, the LBG population was divided into 13 families.”

2.”Finally, the LBG population was divided into families based on the clustering results of male goats and the criterion that individuals with a kinship greater than 0.1 belonged to the same family.”

Point 8: Line 33: Give the names of the breeds.

Response 8: Dear reviewer, We completely agree with your point of view and have followed your suggestions.

Corrected to: ”...with the Nanjiang goat and Qinggoda goat populations...”.

Point 9: Line 38: Give the names of the commercial breeds.

Response 9: Dear reviewer, We completely agree with your point of view and have followed your suggestions.

Corrected to: ”In addition, in a comparative analysis with three commercial breeds, Saanen goat, Boer goat and Angora goat, we also found multiple genes related to physical characteristics (ERG, NRG3, and EDN3, etc.).”

Point 10: Instead of ‘ Finally, we functionally enriched these genes and…’ write ‘Finally, we performed functionally enrichment analysis on these genes and…’

Response 10: Dear reviewer, thank you for your correction.

Corrected to: ”Finally, we performed functionally enrichment analysis on these genes and explored their genetic mechanisms.”

Point 11: Delete ‘In summary, the LBG population has achieved good results in seed conservation and exhibited unique genetic structure and selection characteristics.’. It gives no additional information.

Response 11: Dear reviewer, We completely agree with your point of view and have adopted your suggestion and deleted the sentence.

Point 12: what does ‘disorderly introduction’ mean? Please elaborate.

Response 12: Dear reviewer, we apologize for causing you trouble due to our incorrect description.

The main meaning of the word "disorderly introduction" in the article is that in the past, in order to breed new breeds, people used a large number of Lvliang Black Goats to breed with other goat breeds, which caused a decrease in the number of purebred Lvliang Black Goats. We also realized that this word could not convey what we wanted to express to readers. Therefore, we made appropriate additions here in the article.

The added content is:

“In 1979, the number of LBGs in Shanxi Province accounted for about 40% of the total number of goats in the province. However, due to the extensive use of LBG in breeding with other breeds during the development of new varieties, the number of purebred LBGs has dropped sharply. In a 2009 survey of purebred LBG, only about 440 were found. The LBG was once on the verge of extinction. Therefore, LBG, an important local goat breed resource, has received attention from the country, and an LBG conservation farm was established in 2010 to carry out conservation work.”

Point 13: Line 60: Please delete ‘seed’. Give the reference, where ‘the list of Chinese livestock and poultry breed resources’ can be accessed.
Response 13: Dear reviewer, thank you for your correction. We removed 'seed' as you suggested. The “List of Chinese Livestock and Poultry Breed Resources” is not suitable for citation, so we have changed this content.
Corrected to: ”Therefore, LBG, an important local goat breed resource, has received attention from the country, and an LBG conservation farm was established in 2010 to carry out conservation work.”

Point 14: Line 65: Delete ‘means’. 

Response 14: Dear reviewer, thank you for your correction. We removed 'means' as you suggested.

Point 15: Line 67: Delete ‘Colli used’, Write: ‘…Chip was used to evaluate…’.

Response 15: Dear reviewer, thank you for your correction. Following your comment, we have corrected ‘Colli used’ to “Goat SNP50 Bead Chip was used to evaluate”.

Point 16: Line 71: This part is unnecessary, please delete: ‘providing valuable genetic evidence for in-depth analysis of the domestication process and migration routes of goats’

Response 16: Dear reviewer, thank you for your correction. We followed your advice and deleted the sentence "providing valuable genetic evidence for in-depth analysis of the domestication process and migration routes of goats"

Point 17: Lines 78-83: This MS is focusing on goat. The pig aspect should be removed. If it remains here, why did the authors stop here? Horse, sheep, and cattle references should also be given. That is why I say there is no need to give this example on pigs.
Response 17: Dear reviewer, We completely agree with your point of view and have followed your suggestions. We rewrote the paragraph citing goat-related literature.

Rewritten paragraph: ”A study using SNP chip technology found that most local populations in South Africa have gene exchange with commercial goat populations and have a common genomic ancestor, which is related to the preference of people in the region to breed goat breeds with good production efficiency. This study reminds policymakers to pay attention to the impact of hybridization and emphasizes the importance of protecting the core population of goats [6].”

Point 18: Lines 89-96: References on non-Chinese breeds should also be given.
Response 18: Dear reviewer, We completely agree with your point of view and have followed your suggestions. We rewrote this section and added content about BoHuai goats and Pakistani Teddy goat.

Rewritten paragraph: ”These signals can effectively reveal the germplasm characteristics of a population when the population size is limited or the phenotypic data is incomplete. In the analysis of selection signals of 30 BoHuai goats, key genes related to lipid metabolism and disease resistance were successfully identified [8]. Similarly, analysis of the Pakistani Teddy goat population also identified multiple candidate genes related to important production traits [9].”

Point 19: Line 92: The reference 8 on cattle is raising the same issue, that was mentioned about pigs. This part could be deleted or give non-chinese examples as well.
Response 19: Dear reviewer, We completely agree with your point of view and have followed your suggestions. We rewrote this section and added content about BoHuai goats and Pakistani Teddy goat. For the rewritten paragraph, see Response 18.

Point 20: Line 109: Replace 'conservation’ with ‘core’.
Response 19: Dear reviewer, We completely agree with your point of view and have followed your suggestions. We replace 'conservation' with 'core'.

Point 21: Line 112: Delete ‘dominant’.
Response 21: Dear reviewer, thank you for your correction. We removed 'dominant'.

Point 22: Line 115: Change ‘germplasm characteristics’ to ‘on goat genome’.

Response 22: Dear reviewer, We completely agree with your point of view and have followed your suggestions. We changed ‘germplasm characteristics’ to ‘on goat genome’.

Point 23: Line 120: Delete ‘from’. Give the company name, and the country as well.
Response 23: Dear reviewer, we are deeply sorry for the inconvenience caused to you due to the lack of article information. We have deleted 'from' according to your suggestion. And added the name of the company in the text: "Beijing Kangwei Century Biotechnology Co., Ltd".

Point 24: Line 122: Change ‘standard’ to ‘…(the accepted value to include the sample in the analyses was …) 
Response 24: Dear reviewer, thank you for your correction. We changed 'standard' to '...(the accepted value to include the sample in the analyses was OD260/OD280=1.7-2.1).

Point 25: Line 135: Please change ‘…to the two chip data were…’ to ‘on the two chips were…’
Response 25: Dear reviewer, thank you for your correction. We followed your suggestion and changed the line ‘…to the two chip data were…’ to ‘on the two chips were…’.

Point 26: Line 148: ‘length’ and ‘fragment’: Use plural forms.
Response 26: Dear reviewer, thank you for your correction. We corrected 'length' and 'fragment' to plural forms 'lengths' and 'fragments'.

Point 27: Lines 157-158: Give details, how pedigrees were determined.
Response 27: Dear reviewer, we apologize for the unclear content of the article due to our incorrect description. This study divided families according to the clustering results of male goats and the standard that individuals with a kinship relationship greater than 0.1 were considered to be in the same family. We have added this in the article. See Response 2 and Response 7 for details of the supplementary content.

Point 28: Line 170: There is just one root in one analysis. Please specify, when WG1 and WG2 were used in which migration analyses.
Response 28: Dear reviewer, we apologize for causing you trouble due to our incorrect description. Due to our mistake, we did not point out in the article that the results used in this study were migration analysis of the root WG1. We have corrected it in the article.

Corrected to: “WG1 were used as roots to infer potential migration events in Treemix v1.13 software.”

Point 29: Line 224: 13 Pedigrees are Based on which characteristic of the tree? Please mark the branches, which are thought to be representing the pedigrees, with thicker lines, As for the colours of the samples, it seems, they are not uniformly grouped on a given branch.
Response 29: Dear reviewer, we are sorry for the inconvenience caused to you due to our mistake. This study divided families according to the clustering results of male goats and the standard that individuals with a kinship relationship greater than 0.1 were considered to be in the same family. In Figure 2D, we mainly want to express the family division, and the clustering of all individuals will confuse our results. Therefore, we replaced Figure 2D with the clustering of male goats.

Point 30: Line 227: IBS ranges from 0 to 1. On the Figure I see it is from -1 to 1. Please remedy.
Response 30: Dear reviewer, thank you for your correction, and we sincerely apologize for the inconvenience caused to you due to our mistake. We have redrawn Figure 2B and Figure 2C.

Point 31: Figure 3: It is not easy to distinguish similar colours (additional information: the writer of these rows went through a colour blindness test, without a mistake). Please introduce different forms of markers. E.g.: rectangular, diamond, triangle, etc.  Please avoid to have the same marker type for two similar colours.

Response 31: Dear reviewer, thank you for your valuable suggestions. We have redrawn Figure 3B to better distinguish the populations.

Point 32: Line 362: The value 0.024 is not lower than 0.017. Please rewrite, and describe what the 0.024 value indicates in the term of health and adaptibility comparing them to the other breeds.
Response 32: Dear reviewer, we apologize for the unclear content of the article due to our incorrect description. In this paragraph, we want to explain the level of inbreeding coefficient of LBG population in goat population, and then combine the results of G matrix and IBS matrix to evaluate LBG population in general. We have changed the statement here to make the expression clearer.
Corrected to: “The FROH of the LBG group was 0.024, which was at a relatively low level among goat groups (0.017 ~ 0.086) [30]. At the same time, by further constructing the IBS and G matrices, we found that the kinship and genetic distance between LBG individuals were relatively small. This low inbreeding degree and kinship of the LBG population are conducive to maintaining the long-term survival and reproduction of the LBG population [31].”

Point 33: Line 367: This description is missing from the Materials and methods. Please insert this explanation there as well, with more technical details.

Response 33: Dear reviewer, we apologize for causing you trouble due to our incorrect description. We have supplemented the criteria for dividing families in Materials and Methods. For details of the supplementary content, see Response 2 and Response 7.

We followed your suggestion and added it here.

The added content is: “We divided the LBG breeding population into 13 independent family lines based on the clustering results of male goats and the relationship between individuals to provide a reference for subsequent breeding work.”

Point 34: Line 369: ‘…by national breeding sites.’
Response 34: Dear reviewer, we apologize for causing you trouble due to our incorrect description. The minimum standard in the article comes from China's "Regulations on the Management of Livestock and Poultry Genetic Resources Conservation Farms, Protection Areas and Gene Banks". We realize that this standard does not meet international standards, so we have corrected the content of the article.

Corrected to: "We divided the LBG breeding population into 13 independent family lines based on the clustering results of male goats and the relationship between individuals to provide a reference for subsequent breeding work.”

Point 35: Line 375: Do the authors need the word 'relatively'? If yes, why is the distance called ‘relatively distant’? Examples from other research or from the presented data should be given.
Response 35: Dear reviewer, thank you for your valuable suggestions. We realised that the word 'relatively' was not appropriate here, so we rewrote the paragraph to make it clearer.

Rewritten paragraph: "This study found that Chinese goats can be clearly distinguished from commercial goat populations. Among Chinese goat populations, the LBG population has the closest genetic correlation with the NJ and QG populations, and the genetic clustering is obvious.”

Point 36: Line 383: Please replace ‘my country’ with a less personal term.
Response 36: Dear reviewer, thank you for your correction. We replace ‘my country’ with ‘China’.

Point 37: Line 386: Again, the word 'relatively'. See the comment above given for Line 375.
Response 37: Dear reviewer, thank you for your correction. We deleted the word 'relatively'.

Point 38: Lines 389 and 391: Is ‘may’ the proper word? Isn’t it known, from historical data, that mixing occured? The results presented here supports more than ‘may’. Can you say LBG is a unique population based on the results? Does it support the uniqueness or not?
Response 38: Dear reviewer, we apologize for causing you trouble due to our incorrect description. We removed the word ‘may’.

We did not find any literature on the mixing of the two populations, but historical records show that LBG has a breeding history of nearly a thousand years. This discussion is a reasonable speculation based on the results of this study.

Thank you for your correction. The statement “LBG is a unique population” in the article is not rigorous, so we have modified the content here and replaced ‘unique’ with ‘special’.

Point 39: Line 438: Change ‘In addition, to fully understand the germplasm characteristics of LBG, we…’ to ‘In addition, to further explore the genome of LBG, we…’
Response 39: Dear reviewer, We completely agree with your point of view and have followed your suggestions. We replaced “In addition, to fully understand the germplasm characteristics of LBG, we…” with “In addition, to further explore the genome of LBG, we…”.

Point 40: Line 450: Delete ‘…Darwish's study pointed out that…’
Response 40: Dear reviewer, thank you for your correction. We deleted “…Darwish’s study pointed out that…”.

Point 41: Line 452: ‘What do 'common genes' mean here? Are they common based on what? Or are these the genes found by all the approaches presented in this study? Please clarify.
Response 41: Dear reviewer, we apologize for causing you trouble due to our incorrect description. The 'common genes' in the article refer to genes detected in all methods. We have revised the article according to your suggestions.

Corrected to: “In this study, there were 12 common genes among the genes annotated by the four selection signal methods, “

Point 42: Line 460, first sentence: It is enough to write: ‘Limitations of this study are...’.
Response 42: Dear reviewer, We completely agree with your point of view and have followed your suggestions.

Corrected to: “Limitations of this study include the following: First, ...”

Point 43: Line 463: Has the LBG whole genome sequence been published/reported somewhere? If not, the 'sequencing' word seems to be more appropriate than ‘resequencing’.
Response 43: Dear reviewer, We completely agree with your point of view and have followed your suggestions. We did not find any reports related to the whole genome sequence of LBG. We followed your suggestion and replaced "resequencing" with "sequencing"

Point 44: Line 467: Replace ‘it’ by ‘breed’.
Response 44: Dear reviewer, thank you for your correction. We followed your suggestion and replaced ‘it’ with ‘breed’.

Point 45: Line 468: Instead of ‘distantly related, and the seed conservation effect was good.’ write ‘distantly related.The conservation effort was effective.’
Response 45: Dear reviewer, We completely agree with your point of view and have followed your suggestions. We replaced “distantly related, and the seed conservation effect was good.” with “distantly related. The conservation effort was effective.”

Point 46: Lines 468-469: Delete ‘In addition, LBG has a unique genetic structure and selection characteristics.’.
Response 46: Dear reviewer, thank you for your correction. We deleted ‘In addition, LBG has a unique genetic structure and selection characteristics.’

Point 47: Line 470: Instead of ‘identified selected genes’ it is more appriate to write ‘identified genes under selection’.
Response 47: Dear reviewer, We completely agree with your point of view and have followed your suggestions. We replaced ‘identified selected genes’ with ‘identified genes under selection’.

Point 48: Line 471: Delete ‘superior’.

Response 48: Dear reviewer, thank you for your correction. We deleted 'superior'.

Point 49: Lines 471, 472: Delete ‘study further analyzed the germplasm characteristics and genetic mechanisms of LBG,’.
Response 49: Dear reviewer, thank you for your correction. We deleted ‘study further analyzed the germplasm characteristics and genetic mechanisms of LBG,’.

Point 50: Line 472: Insert a ‘to’ before ‘support’, delete ‘for’. Replace ‘…protection and breeding of LBG,’ with ‘...protection and improvement of LBG.
Response 50: Dear reviewer, We completely agree with your point of view and have followed your suggestions.

Corrected to: “This provided data to support the future protection and improvement of LBG,”

Round 2

Reviewer 2 Report

Comments and Suggestions for Authors

The new analyses and changes in the wording and information provided to the paper have substantially improved the work, so I propose its acceptance for publication.